# Fundamental Chemistry of Essential Oils and Volatile Organic Compounds, Methods of Analysis and Authentication

**DOI:** 10.3390/plants11060789

**Published:** 2022-03-16

**Authors:** Nicholas J. Sadgrove, Guillermo F. Padilla-González, Methee Phumthum

**Affiliations:** 1Royal Botanic Gardens, Kew, Kew Green, Richmond TW9 3DS, UK; n.sadgrove@kew.org (N.J.S.); f.padilla@kew.org (G.F.P.-G.); 2Department of Pharmaceutical Botany, Faculty of Pharmacy, Mahidol University, Bangkok 10400, Thailand

**Keywords:** introduction, beginners, basic chemistry, learn

## Abstract

The current text provides a comprehensive introduction to essential oils, their biosynthesis, naming, analysis, and chemistry. Importantly, this text quickly brings the reader up to a level of competence in the authentication of essential oils and their components. It gives detailed descriptions of enantiomers and other forms of stereoisomers relevant to the study of natural volatiles and essential oils. The text also describes GC-MS work and provides tips on rapid calculation of arithmetic indices, how to interpret suggested names from the NIST mass spectral library, and what additional efforts are required to validate essential oils and defeat sophisticated adulteration tactics. In brief, essential oils are mixtures of volatile organic compounds that were driven out of the raw plant material in distillation, condensed into an oil that is strongly aroma emitting, and collected in a vessel as the top layer (uncommonly bottom layer) of two phase separated liquids: oil and water. Essential oils commonly include components derived from two biosynthetic groups, being terpenes (monoterpenes, sesquiterpenes and their derivatives) and phenylpropanoids (aromatic ring with a propene tail). The current text provides details of how terpenes and phenylpropanoids are further categorised according to their parent skeleton, then recognised by the character of oxidation, which may be from oxygen, nitrogen, or sulphur, or the presence/absence of a double bond. The essential oil’s science niche is an epicentre of individuals from diverse backgrounds, such as aromatherapy, pharmacy, synthetic and analytical chemistry, or the hobbyist. To make the science more accessible to the curious student or researcher, it was necessary to write this fundamentals-level introduction to the chemistry of essential oils (i.e., organic chemistry in the context of essential oils), which is herein presented as a comprehensive and accessible overview. Lastly, the current review constitutes the only resource that highlights common errors and explains in simplistic detail how to correctly interpret GC-MS data then accurately present the respective chemical information to the wider scientific audience. Therefore, detailed study of the contents herein will equip the individual with prerequisite knowledge necessary to effectively analyse an essential oil and make qualified judgement on its authenticity.

## 1. Introduction

Essential oils and natural volatiles are one of industry’s most successful commodities, since they are used as flavours [1], in lotions or shampoos as fragrances and for skin and hair rejuvenation [2,3], as perfumes, in candles, soaps, liquids that sterilise, and in conventional or medical aromatherapy [4]. This widespread demand has meant that significant profits are to be made by providing essential oils or their components at the most competitive price. This creates the motive to explore short-cuts in supply [5], to produce them by synthesis or in a bioreactor, or to use counterfeits or adulterants. 

In general, the consumer wants their essential oils to be derived from plants as in the traditional method. Some consumers will tolerate synthetic essential oils, if they are produced in a bioreactor using genetically modified yeasts [6] or plant cell cultures [7], rather than by traditional synthesis that uses potentially dangerous chemicals. 

Most consumers of essential oils are concerned about the authenticity of the product. This concern is justified because it is common for fixed oils, such as sunflower, mustard or canola oil, to be scented with a few drops of a fragrance, then sold as a ‘pure’ essential oil. Furthermore, aromatic plants are sometimes extracted into fixed oils, then advertised as an ‘essential oil’, which implies a higher quality product [5]. Fortunately, consumers are becoming more aware of the difference between an essential oil and an extract, concrete, absolute or enfleurage perfume [4,8].

Normally, an essential oil is a mixture of volatile organic compounds that were produced by a type of distillation, such as hydrodistillation, steam distillation, or microwave assisted dry distillation [9]. It is wrong to say that aromatic plants contain essential oils because the essential oil is a product of distillation. However, it is correct to say that aromatic plants have the ingredients and precursors that are required to make an essential oil, but they are usually called essential oil components [4]. 

According to the international standards organisation, there is one exception to this rule. Oil produced by the mechanical pressing of citrus fruit rinds, such as Bergamot, can be called an essential oil [10], which may be due to an established tradition that was ‘grandfathered’ into the current standard. Otherwise, if not from citrus, pressed oils are correctly identified as ‘fixed’ or ‘expressed’ oils. Extracts are correctly identified as concretes (hexane extracts), fat extracts, absolutes (ethanol extract of concretes), and enfleurage perfumes. 

There is a pronounced difference between essential oils and extracts, because essential oils are composed of a narrower range of volatile molecules compared to extracts. However, in cosmetic or therapeutic applications, there is a place for both types, i.e., bergamot extracts contain UV sensitising coumarins, such as bergapten, which will cause blisters on candidates if they are in direct sunlight when using bergamot oil. In this scenario, the hydrodistilled bergamot oil is safer because there are no coumarins in the essential oil. 

Alternatively, aromatic extracts that include fixed components, such as flavonoids or alkaloids, generally produce synergistic antimicrobial effects [11,12,13]. In cases of antimicrobial synergy between fixed and essential oil components, the same antimicrobial potency cannot be produced using only the hydrodistilled oils.

Essential oils are mixtures of volatile organic compounds that were either biosynthesised in specialised plant cells or produced as an artefact of the distillation process from chemical precursors in the plant material. Examples of heat derived artefacts include geijerene in essential oils of *Geijera parviflora* L., which is derived from the precursor pregeijerene [14], elemol, which is produced from hedycaryol [15,16], the eudesmol isomers, which are produced from cryptomeridiol [17], and spathulenol, which is produced from bicyclogermacrene but is also naturally occurring [18]. 

The production of artefacts in distillation means that there are ingredients in essential oils that are not naturally occurring in the raw plant material. Sometimes the artefacts are present naturally in plant material, but their concentration increases by conversion of their chemical precursors (i.e., cryptomeridiol and bicyclogermacrene). Hence, essential oils are made from natural ingredients, but not all essential oil components are natural per se.

The etymology of essential oil is in the Latin name *quinta essentia*, which means the 5th element. Because the 5th element is the ‘spirit’ then an essential oil is a spirit that has been captured from the air or condensed from distillation [19]. However, volatile organic components are not evaporated in the same sense as a spirit because their individual boiling points are not met in distillation. The higher temperature used in steam or hydrodistillation increases the vapour pressures of the volatile organic compounds, then the steam from the boiling water pushes the hot fragrant vapour towards the condenser where they are cooled. The sudden change in vapour pressure causes precipitation, returning most of the compounds and the water to the liquid phase. 

The discipline of essential oils attracts individuals from diverse backgrounds, such as aromatherapy, pharmacy, synthetic and analytic chemistry, just to name a few. For this reason, it is necessary to provide a concise fundamentals-level introduction to the chemistry of essential oils (i.e., organic chemistry in the context of essential oils) to empower non-chemist researchers, enthusiasts, and students with a comprehensive and accessible overview. Because of the numbers of students who are learning analysis for the first time, it is also necessary to give an overview of the methods used to chemically characterise essential oils. Lastly, this text intends to empower people to read between the lines when assessing the quality of products available in the marketplace, or to further scrutinise the quality of essential oils produced by themselves. 

## 2. Chemical Classification of Essential Oil Components

Essential oil components are chemically classified according to four conditions: (1) primary biosynthetic origin (Figure 1, Figure 2, Figure 3 and Figure 4); (2) size or number of carbon atoms; (3) parent backbone or ‘skeleton’, and (4) character of oxidation by electronegative atoms, i.e., other atoms larger than carbon, such as oxygen, nitrogen, or sulphur (Figure 5 for character of oxidation). There are four major biosynthetic pathways that produce three biosynthetic groups of essential oil components, either terpenes (Figure 1, Figure 2 and Figure 3), phenylpropanoids or isothiocyanates (Figure 1 and Figure 4). They derive from the mevalonate (Figure 2) and methylerythritol phosphate (Figure 3) [20], shikimate (Figure 4) [21], and glucosinolate [22] biosynthetic pathways, respectively.

### 2.1. Biosynthesis of Terpenes

The precursors of isoprene, isopentenyl, and dimethylallyl diphosphate are produced by two distinct biochemical pathways, which are the mevalonic acid (MVA: Figure 2) and 2C-methyl-D-erythritol-4-phosphate (MEP: Figure 3) pathways. The MVA pathway is active in all eukaryotes, but the MEP pathway is restricted to a selection of organisms, including algae and higher plants [23].

The MVA pathway starts with the condensation of small acetyl groups (43 Da) cleaved from acetyl coenzyme A, a much larger molecule (809.6 Da). Condensation of acetyl groups from two carbon units to produce diketide chains justifies why natural alkanes are formed with even carbon numbers. However, with terpene synthesis the condensation stops at six carbons. After the formation of mevalonate 5-phosphate or -diphosphate, the enzymatic removal of the ionised acid group eliminates a carbon, reducing the intermediate to a 2-methyl five carbon unit, which is the fundamental unit of all terpenes [23].

The MEP pathway is fundamentally different to the MVA pathway. The MEP pathway starts with condensation of pyruvate and glyceraldehyde derivatives to produce a polyhydroxylated five carbon chain (1-deoxy-*D*-xylulose). Thereafter, an enzyme that belongs to the reductoisomerase group converts the five-carbon chain into a methylated four carbon chain, which represents a significant step in the formation of the isoprene used to construct terpenes [23].

In higher plants, the MVA and MEP pathways occur in different compartments of photosynthetic tissue. The MVA pathway is active in the cytosol, whereas the MEP pathway is active in the plastids. Lastly, the MVA pathway produces sesquiterpenes, sterols, and triterpenes, whereas the MEP pathway produces monoterpenes, diterpenes, and tetraterpenes [23].

Several studies of phenotypic plasticity of essential oils have inadvertently identified modulation of these pathways in response to abiotic factors. Expression differences between the MVA and MEP pathways often manifests as differences in expression between sesquiterpenes (MVA) and monoterpenes (MEP). For example, chemophenetic studies of essential oils in the genus *Phebalium* frequently identify taxa that normally produce sesquiterpenoid essential oil profiles but have members that express higher yields of essential oils that are dominated by monoterpenes [15,24]. In such cases, the expression of sesquiterpenes has not changed, but the higher expression of monoterpenes causes the sesquiterpene components to be diluted and underrepresented in the chemical profile. A similar effect occurs in the *Prostanthera lasianthos* heterogeneous species aggregate, between the monoterpenes 1,8-cineole (*P*. sp. Wollomombi Gorge) or α-pinene (*P. eungella* B.J.Conn and K.M.Proft) and sesquiterpenes [25]. 

### 2.2. Terpenes and Meroterpenes

As previously mentioned, terpenes are constructed out of isoprene, which is a five-carbon branched chain. Hence, terpene sizes are quantised by a factor of five carbons. Monoterpenes (single terpene) are made of 10 carbons (two isoprenes) and sesquiterpenes of 15 (three isoprenes) (Figure 1). Although it is less common, diterpenes (20 carbons; four isoprenes) are also present in essential oils. 

An example of a diterpenoid essential oil is the abietane and halimane mixture obtained from the leaves of the Pretoria variant of *Croton gratissimus* Burch [26]. Gratissimone is a previously undescribed abietane that stood out as a major component of the essential oil, together with gratissihalimanoic ester. The latter is an example of an ester. In this regard, terpenes can have carbons added or removed, such as esters (methyl ester, +one carbon), acetates (acetate ester, +two carbons), or norterpenes, which have carbons removed (i.e., C19-norterpene). However, gratissihalimanoic ester has only 20 carbons in total, despite being an ester. This is because it is also a C19-norterpene, meaning it has also lost a carbon, which is probably a consequence of the instilling of the double bond at the position of attachment of the furan moiety to the bicyclic ring (Figure 6). 

In the case of gratissihalimanoic ester, a carbon is added by methyl esterification of the acid group, making it an ester. The esterification is the reason it was present in the essential oil. Without esterification its vapour pressure would be too low to be driven into the essential oil during distillation. Similarly, by following an esterification protocol, many fixed diterpene acids, such as crotohalimaneic acid (Figure 6), can be converted to esters and studied by GC-MS [27].

Terpenes can also combine with phenylpropanoids or phloroglucinols to become ‘meroterpenes’, such as osthole, which is a prenyl coumarin that is often detected in essential oils [28]. Nevertheless, there are many examples of meroterpenes in the genera *Peperomia* [29] and *Psoralea* [30], such as bakuchiol (Figure 7) that are light enough and have polar head space that is low enough to enable collection by hydrodistillation, so it is just a matter of time before such essential oils will be reported in the published literature.

### 2.3. Biosynthesis of Phenylpropanoids

The shikimate pathway is responsible for the biosynthesis of all phenylpropanoids. It is predominantly active in the plastids, which is where several of the primary enzymes originate [31]. The shikimate pathway produces many compound classes, such as phenols, phloroglucinols, lignin, flavonoids, amino acids, and so on. There are several significant steps that direct synthesis to the phenylpropenes. An example of such a step is the conversion of phenylalanine or tyrosine, modulated by aromatic amino acid lyases. The subsequent cinnamate is then converted to p-coumaroyl by the enzyme ‘cinnamate 4-hydroxylase’, then coenzyme A is esterified to the acid group by 4-coumaroyl CoA ligase. 

This represents a significant turning point in biosynthesis, as p-coumaroyl CoA is a common branching point. Other important steps come after p-coumaroyl CoA, particularly to direct the derivisation toward phenylpropene synthesis as opposed to flavonoids, stilbenes, or related derivatives. The diversity of the phenylpropanoid family of volatile organic compounds is thereafter controlled by the various genus-specific genes that encode for oxygenases, reductases, and transferases that create functional groups responsible for the observed diversity [31].

### 2.4. Phloroglucinols and Phenylpropanoids 

Volatile phloroglucinols and phenylpropanoids are typically made from the same phenylalanine or tyrosine precursor and are not characterized by ‘units’ in the same way as terpenes. 

A phloroglucinol is built around a central ring that is aromatic in the chemical sense (delocalised electrons), i.e., the core of a phloroglucinol is a trihydroxylated benzene ring. The hydroxyl groups are alternating, i.e., on every second carbon around the ring. Naturally occurring phloroglucinols are often substituted on the non-hydroxylated carbons, often with an aldehyde or acetyl group. An example of a phloroglucinol dominated essential oil (or solid) is derived from the Australian species *Geijera parviflora* Lindl., which is dominated by xanthoxylin (Figure 7) [14].

Phenylpropanoids are more common in essential oils than phloroglucinols. Like phloroglucinols, phenylpropanoids are aromatic compounds in the chemical sense (delocalised electrons, benzene moiety) that are not classified according to size but rather the character of oxidation (the attachment of relatively electronegative atoms), which includes alcohols, phenols, aldehydes, methoxy, and methylenedioxy variants (Figure 5). 

Two very well-known phenylpropanoids are anethole (*trans*-anethole) and safrole. *Trans*-anethole, or simply anethole, is the molecule responsible for the flavour of liquorice or the liquor sambuca [32]. Although *trans*-anethole is an accepted flavour ingredient in industry, safrole is avoided because a study in the 1960s demonstrated that safrole may be associated with hepatotoxicity and liver cancers in mice [33], and later it was determined that a metabolite of safrole was possibly the carcinogenic principle [34]. Although this outcome was never demonstrated in humans, safrole is no longer used as a flavour ingredient, but small amounts are still present in common spices [35]. Another significant phenylpropanoid is elemicin, which is the psychoactive ingredient in nutmeg (*Myristica fragrans* Houtt) [36]. Due to the psychotropic action of the phenylpropanoids in nutmeg, the herb has become the basis of a systemic problem of ‘nutmeg’ abuse [37]. A major concern of nutmeg abuse is the potential for toxic effects on the liver. 

### 2.5. Parent ‘Skeleton’ and Character of Oxidation

Monoterpenes, sesquiterpenes, and diterpenes can be cyclic or acyclic and are then classified according to the parent skeleton (i.e., terpinene or limonene). Details of the parent skeleton of natural products can be found in the Dictionary of Natural Products [38,39], but there are other works that provide comprehensive summaries of specific groups, such as the Atlas of Sesquiterpene Hydrocarbons [40]. 

Parent groups, such as abietanes, cadinanes, or germacranes, are further recognised by the character of oxidation, which may include no oxidation (simple terpenes), alcohols, phenols, aldehydes, methoxylates, acids or acid esters, ethers or oxides (pyrans, furans), ketones, methylene dioxides, lactones and very rarely, coumarins (Figure 5). Peroxides are also detected in essential oils but are usually a product of oxidation processes during aging, and they are an intermediate as the oil goes rancid (in fixed oils mainly). Naturally occurring peroxides also occur, however, they are sometimes degraded by heat. Artemisinin is an example of a naturally occurring peroxide from *Artemisia annua* L., but its vapour pressure is not high enough for it to be driven into the condenser in hydrodistillation [41] and it may decompose at higher temperatures to produce volatile derivatives. Alternatively, ascaridole (Figure 7) from a few species in *Chenopodium* is volatile enough to be captured in an essential oil [42]. 

In the jargon of essential oil components there is a convention of modifying suffixes to add information about chemical class, by combining two words into one. For example, monoterpene alcohols are shortened to ‘monoterpenols’ [43] and monoterpene ketones are shortened to ‘monoterpenones’, and so on. 

A common monoterpenol is menthol, which is the principal flavour compound of mint. A common monoterpenone is menthone, which is also present in mint. Both compounds are related according to the parent compound, menthane, but they are individualised according to the nature of oxidation. On the converse, by reducing these compounds, leading to the removal of the oxygen from the structures, they are returned to the parent form as menthane. Furthermore, limonene is also a menthane that has acquired a double bond at two sections of the molecule. Similar to menthol and menthone, limonene can be returned to the parent structure menthane, also by a reductive process, but in this case, it requires hydrogenation to remove the double bonds. Other examples of parent compounds and their derivatives include: pinane, which is the parent of α- and β-pinene; abietane, which is the parent of gratissimone [26] and pisiferal [44]; humulane, which is the parent of humulene; caryophyllene, which is the parent of caryophyllene and caryophyllene oxide; cadinane, which is the parent of cadinene, muurolol, and cubenol; and so on. 

### 2.6. Less Common and Rare Components

The Australian continent is home to several species that express rare or previously undescribed volatile organic compounds, whose biosynthetic origin remains a mystery for now [45]. Another area of endemic essential oils is South Africa [8]. An example of a rare South African methyl-thio-ester called nudaic ester (Figure 7) was discovered in the essential oil distilled from rhizomes of *Annesorhiza nuda* [46]. The biosynthetic origin of this 2-butyl ester is unclear. The same could be said for the Australian 1-methylbutyl, butanoic ester from *Prostanthera sp. aff. lasianthos* [25]. 

Other components have well known biosynthetic origins but are not expected in an essential oil. For example, another rare essential oil is the xanthoxylin dominated chemotype (syn. phloroacetophenone dimethyl ether) of the Australian shrub *Geijera parviflora* [14,47]. From the same species another chemotype was found to have traces of the coumarin osthole [14,48], but this required higher temperatures and longer distillations. Other coumarins that appear in essential oils accelerate UV damage to the dermis, such as the photogenotoxic psoralens, bergapten, and xanthotoxin [49]. They can be produced by hydrodistillation of many citrus species, but most citrus ‘essential oils’ in the market are cold pressed (not distilled), which affords a coumarin or polymethoxylated flavonoids-rich extract [50]. 

Other less common compounds are isothiocyanates (Figure 1), phloroglucinols, and short chain fatty acid esters (Figure 7), which are biosynthesised by the glucosinolate, shikimic acid, and glycolytic pathway, respectively. These components that are uncommon in essential oils are usually found in onion, garlic, cruciferous vegetables, and fruits. The isothiocyanates and thiocyanates are exclusively found in the order Brassicales, usually bound to a sugar moiety at the sulphur end, and are released when the vegetative parts are damaged, either by chewing or by crushing, which removes the compartmental barrier between the compound and the enzyme myrosinase [51]. This effect is known to result in a sudden release of flavours when the enzyme cleaves off the glucose moiety, converting the glucosinolate to the isothiocyanate [52], which may include isothiocyanate volatiles such as those depicted in Figure 1.

### 2.7. Colour and Viscosity of Essential Oils

In addition to the enfleurage effect that occurs in combinations of volatile organic compounds, significant in olfactometry and aesthetics, combinations also confer different viscosities and colours. For example, the eudesmol-rich essential oil from species in the genus *Leptospermum* [45,53] is as viscous as honey, whereas the eudesmol content of *Eucalyptus nova-anglica* H.Deane and Maiden is so high that the oil is sometimes a solid, similar to wax in appearance, or like a block of camphor [54]. 

Furthermore, in several instances sesquiterpene dominated essential oils remain fluid while they are warmed by the hydrosol, but after they have been placed into a vial they stiffen and become like toffee or as thick as molasses. In such instances the condenser is coated with a sesquiterpene-rich layer of the less volatile fraction of the essential oil. This is common when distilling essential oils from a chemotype of *Prostanthera lanceolata* Domin that is rich in cis-dihydroagarofuran and kessane [55]. To recover the sesquiterpenes, the condenser is washed with hexane or diethyl ether. Use of high-vapour pressure solvents enables rapid evaporation of the solvent with minimal fractionation to the recovered essential oil. 

Alternatively, the prostantherol, maaliol, and globulol-rich essential oils (Figure 8) from species in the genus *Prostanthera* produce a thick essential oil that will later grow crystals during storage, which are long needle-like prisms that sink to the bottom of the oil. These crystals can sometimes occupy >50% the volume of the essential oils [55]. A similar outcome is derived from the guaiol-rich essential oils from the timber of species in the genus *Callitris* (Cuppressaceae) [27]. Alternatively, hydrodistillation of the xanthoxylin chemotype of *G. parviflora* will cause the condenser to become blocked with a white paraffin-type granular to amorphous solid (xanthoxylin, syn. phloroacetophenone dimethyl ether) [14,48]. In this case it is not right to call it an oil, since it is a solid, nevertheless, it is an ‘essential solid’.

Many of the volatile organic compounds that become amorphous solids take on a white or translucent white appearance. Those components that become crystals are normally transparent, but they are usually in a liquid that has a yellow appearance, in cases observed by the authors. 

Most essential oils are either completely transparent or pale yellow. Sometimes the yellow colour is caused by linalool, or a yellow compound that represents a major portion of the essential oil, however, there are cases where the colour derives from a trace component that is <0.5%. This happened when the genifuranal-rich essential oils were produced from *Eremophila longifolia* F.Muell. It was initially believed that genifuranal was canary yellow, but when it was purified it became colourless [56,57]. 

Only once have the authors observed a red essential oil, but it acquired the red colour while in storage. The essential oil was produced from *Eremophila alternifolia* R.Br., and the red component was never identified. However, the chemical composition of the oil was dominated by myoporone, dehydromyoporone, and myomontanone [18]. 

Coloured essential oil components are usually a consequence of the conjugation of double bonds, i.e., three or more double bonds in alternating sequence. The more efficient the conjugation, the stronger the colour. For example, Figure 9 depicts coloured essential oil components, and the components with the strongest colour (guaiazulene and chamazulene) have five conjugated double bonds. 

A green essential oil is produced from a chemotype of *G. parviflora* that is currently being called ‘green lavender’. The lavender note comes from the presence of linalool, which is a component that is abundant in lavender essential oils. The green colour derives from two components, pregeijerene and geijerene (Figure 9). The relative quantities of both components are determined by the preparative steps before distillation. 

Pregeijerene is a blue-green compound that is heat labile, so it is converted to geijerene, a green compound, in distillation. However, when the leaves are completely pulverised (using liquid nitrogen), freeing the essential oil components from leaf tissues, the pregeijerene is more likely to survive the hydrodistillation. This is because the distillation time is shortened by freeing the volatile organic compounds from their storage compartments in leaf tissue. Compared to the lighter green colour of geijerene, pregeijerene is a blue-green, which makes the oil darker [14]. 

There are several royal navy-blue essential oils. For example, in South Africa there is a blue essential oil produced from Cape Chamomile (*Eriocephalus tenuifolius* DC.) [58]. The blue colour derives from chamazulene, which only needs to be present in trace quantities to make the oil appear blue. Hydrodistillation produces chamazulene as a byproduct of the heat driven derivatisation of matricin (Figure 9), meaning it is not a naturally occurring compound [59]. Both matricin and chamazulene may contribute to the anti-fever affects achieved in anecdotal accounts [60], because anti-inflammatory compounds are known to confer antipyretic effects by reducing inflammation in the hypothalamus [61].

The oil from Cape Chamomile is closely related to the blue oil from leaves of *Pentzia punctata* Harv. [62], also from the family Asteraceae. A closely related species, *Pentzia incana* (Thunb.) Kuntze, is chemically similar to *P. punctata*, but it does not include the chamazulene, so the oils are not blue [63]. Both species are chemically related to *Artemisia arborescens* L., which also produces this chamazulene rich oil [64]. 

However, chamazulene is not the only compound that makes blue essential oils. In Australia there is a species that is not recognised internationally, *Callitris intratropica* R.T.Baker and H.G.Sm., currently only accepted under the synonym *C. columellaris* F.Muell, which produces a blue essential oil from the timber. The blue component is guaiazulene [65], which is not produced by any of the other species currently regarded as synonymous with *C. columellaris*. 

## 3. Stereochemistry and Isomerism in Essential Oils

Before the development of modern chromatographic techniques, it was common for chemists to fail to separate mixtures of closely related isomers or mistake a mixture of isomers as a single compound. Isomers are molecules with the same molecular formula, such as the simple monoterpene, which has 10 carbons and several hydrogens, with a hydrogen number that reflects the number of double bonds and cyclisation (number of rings) in the molecule. Hence, the molecular formula of limonene, which is C_10_H_16_, can represent thousands of isomers. These ‘constitutional’ isomers are not very similar, so they are usually only recognised when it is contextually appropriate, i.e., when they are similar enough to be grouped together, but differ by the atom-to-atom bonds. For example, compounds that have the same atom to atom connectivity but are different by the position of a double bond, such as α- and β-pinene, are constitutional isomers. 

Another example is the ‘regioisomer’, which is a type of constitutional isomer that has its functional groups attached at different positions, such as 3-methoxy-4,5-hydroxybenzoic acid, compared to 4-methoxy-3,5-hydroxybenzoic acid. In this example, the methoxy group is at position 3 in the former, and position 4 in the latter. Because these molecules are so closely related in terms of structure, it is relevant to acknowledge these as constitutional isomers.

However, compared to constitutional isomers, stereoisomers are more closely related because they have the same molecular formula, as in constitutional isomers, but they also have the same attachment between atoms, including the position of double bonds, but the atoms in bonds at a ‘stereo-centre’ are pointing in different directions. Not all molecules have ‘stereo-centres’ or ‘chiral centres’ but for those molecules that do, they are called stereoisomers, and the two categories of stereoisomer are diastereomer and enantiomer (Figure 10).

### 3.1. Diastereomers 

Species such as *Pelargonium crispum* [66], *Cymbopogon citratus,* and *Citrus*
*×*
*limon* [67], are rich in a lemon smelling mixture of diastereomers called citral. The name citral (syn. lemonal) was first coined by Ernest Guenther in 1948, which was recognised as responsible for the lemon aroma of *Citrus ×*
*limon* [68,69]. 

For more than 10 years it was believed that citral was a single terpene aldehyde [67]. It is unclear when the realisation that this was wrong occurred, but the first separation and structural confirmation of both isomers in the mixture was completed and published in 1964 [70] using fractional distillation for separation and nuclear magnetic resonance spectroscopy (NMR) for structural elucidation. At the time the two isomers were called ‘citral a’ (*trans*-citral) and ‘citral b’ (*cis*-citral), but today they are known as geranial (α-citral) and neral (β-citral). They are *cis* and *trans* diastereomers (Figure 11), which is a specific type of stereoisomer. 

These *cis* and *trans* diastereomers represent one subcategory of diastereomer, which is caused by double bond geometry. Double bond diastereomers, such as citral a and citral b, are denoted with *Z*-(*cis*) and *E*-(*trans*) prefixes to describe the position of priority groups around a double bond. In this case *Z*- or cis denotes priority groups that are positioned directly opposite each other, and *E*- or trans denotes positions away from each other (Figure 12). As mentioned in the discussion in Section 3.3, priority groups are defined according to the Cahn–Ingold–Prelog priority rules which rank according to atomic number, i.e., in the order of O > N > C > H. 

### 3.2. Enantiomers (Chirality)

Stereoisomers are also defined in terms of ‘chirality’ which uses a prefix, such as *l*- or *d*- that is synonymous with (−) or (+), respectively, to acknowledge one of two possible spatial configurations of atoms in the molecule, resulting in a mirror image version of the molecule (Figure 12 and Figure 13). These chiral pairs are called enantiomers and are described using the analogy of left and right hands that cannot be superimposed: the Greek etymology of chiral is ‘kheir’ meaning hand (Figure 13). Chirality is caused by a simple difference in the position of substituents attached at a single carbon centre, called the ‘chiral carbon’. A chiral carbon has four different substituents attached, creating asymmetry (Figure 13). If more than one chiral carbon is present in the molecule then all chiral carbons need to be the mirror image of its enantiomer, otherwise with different spatial configuration(s) at n − 1 chiral centre(s) the molecule becomes a diastereomer. 

The difference between an enantiomer and diastereomer is as follows: in a chiral pair of enantiomers, all chiral centres of the one enantiomer are spatially different to its chiral equivalent, and enantiomers can have one or more chiral centres; alternatively, a pair of diastereomers must have two or more chiral centres where at least one chiral centre is spatially the same between the pair (Figure 11 and Figure 12). 

### 3.3. Fundamentals of Chirality (Enantiomers)

Enantiomers rotate plane polarised light in opposite directions. Rotation of light to the left is represented by a negative (−) symbol, or by *l*- (lower case) meaning laevorotatory from Greek, or left rotation in English. On the converse, movement to the right corresponds to a positive (+) symbol, or *d*-, meaning dextrorotatory from Greek, a synonym for right rotation in English. 

The lower-case descriptors *d*- and *l*- are often confused with uppercase descriptors *D*- and *L*- of the Fischer–Rosanoff convention. These are only used for sugars, carbohydrates, or amino acids, and are not related to the direction of rotation of plane polarised light. That is why sugars are often given two prefixes in denoting stereochemistry, i.e., *D*-(+)-glucose or L-(+)-fructose. These *D*- and *L*- descriptors refer to the rotation of priority groups at just one of the molecule’s carbons, which corresponds to the chiral centre farthest from the keto group on a reducing sugar, or the corresponding carbon in a non-reducing sugar (pyranose). 

Furthermore, when NIST library searches suggest ‘*D*-limonene’ or ‘*D*-camphor’ it is wrong on two counts. If a chiral column was not used in GC-MS analysis, then it is not possible to determine which enantiomer a chiral compound corresponds to using mass spectral data. The NIST library has a 50% chance of getting the chiral descriptor right, because the probability is random, it is just a guess. In which case, the *D*- or *L*- (or ‘*d*-’ or ‘*l*-’) should be removed from the compound’s name, and the compound identities should be corrected to ‘limonene’ and ‘camphor’, until polarimetry work is undertaken, or a chiral column is fitted to the GC-MS.

If a chiral column is used, information about retention time and relative elution order is used to make the chiral assignment. In such cases, it is important to correct the ‘*D*-’ to the lower case ‘*d*-’, i.e., ‘*d*-limonene’ and ‘*d*-camphor’. Remember that the upper-case letters are used for sugars only.

For an absolute description of the configuration of atoms around a chiral carbon, an analogy of a clock is used. In this analogy, movement of the clock’s hand to the left of 12 corresponds to the anticlockwise direction, and to the right corresponds to the clockwise direction. Similarly, if a chiral centre is oriented in space so that the lowest priority atom is facing away from the observer, then a circle is drawn to pass through each of the three remaining atoms in order of priority, the anticlockwise direction is labelled as ‘S’ and the clockwise direction is ‘R’ (Figure 12). 

For information on classification of priority groups see Cahn–Ingold–Prelog priority rules. In brief, a hydrogen is the lowest priority atom, then priority is given to higher mass elements (higher atomic number). If masses are equivalent, then the number of bonds (single, double, or triple) to the neighbouring atoms (beta atoms) defines priority, which is given to higher numbers of bonds. If all have equivalent bonds then the next bonding element is used to determine priority following the same rules, and so on. If two of the atoms are determined as equivalent, with no priority to either, even at several bonds away from the carbon centre in question, then it is not a chiral centre. 

The *S* and *R* descriptors derive from the Latin words sinister and rectus for left and right (or straight), respectively. It is a common misconception that *S*- and *R-* descriptors denote the direction of rotation of plane polarised light, however, this is not true. They are independent of *d*- and *l*- or (+) and (−). For example, *S*-(−)-α-pinene and *S*-(+)-carvone rotate plane polarised light in opposite directions yet both use the *S*- descriptor to define the order of bonds around the chiral centre. 

The effects of chirality are also significant in the context of biological effects [71], particularly due to binding with proteins: imagine a left hand squeezing into a right-handed glove; the hand being the molecule and the glove being the protein or enzyme. Furthermore, the human nose is easily capable of perceiving an olfactory difference between enantiomers. This has been common knowledge for over 50 years. For example, in 1971 a blind study was published on perception of odour differences between the two enantiomers of carvone. The panel independently recognised (+)-carvone as ‘caraway-like’ and (−)-carvone as comparable to spearmint [72]. 

Furthermore, the use of α-, β-, γ-, and δ- (alpha, beta, gamma, and delta, respectively) are for chemical species with the same carbon–carbon connectivity (constitutional isomers) but with different positioning of a double bond, i.e., α-pinene vs. β-pinene (Figure 14), or different configurations of chiral centres leading to epimers (but not enantiomers), i.e., α-thujone vs. β-thujone. Epimers are diastereomers that differ by the position of only one out of two or more chiral centres. Although the denotation of epimer is more commonly used in the language for carbohydrates, epimers are commonly defined in natural products. For example, when epimers are recognised, they are sometimes given entirely different names altogether, such as ledol and globulol. In another example, the furanosesquiterpenes ngaione and epingaione are distinguished in their vernacular names as epimers [18].

Even when there is no double bond in the molecule, epimers can also be defined as ‘cis’ and ‘trans’ to indicate the relative positions of two priority groups attached at two different stereocenters, as illustrated in Figure 11 and Figure 12. Alternatively, they can be distinguished using the prefix ‘iso’, i.e., menthone vs. isomenthone (Figure 14). The cis epimer applies to isomenthone (2S,5S and 2R,5R), having the two methyl/methylene carbons (axial groups) pointing in the same direction in space. The trans epimers are known as menthone ((−)-2S,5R and (+)-2R,5S). Inversion of just one chiral centre, i.e., from 2S,5R to 2S,5S, creates the epimer, isomenthone (Figure 14). However, inversion of both chiral centres, i.e., from (−)-2S,5R to (+)-2R,5S, creates the positive enantiomer. 

Another example could be made of thujone, which has three chiral centres (Figure 14), with two known epimers (α- and β-thujone) that are only naturally known by the one enantiomer. Unlike the menthones, more isomers and enantiomers of thujone are possible, but they are not generated in biosynthesis. The stereochemistry of natural products is limited by their biosynthesis, whereas lab-based synthesis can generate a greater number of possibilities. Hence, sometimes natural enantiomers are not known, and this can be informative when assessing the authenticity of essential oils. Racemic mixtures (equimolar mixtures of both enantiomers) are often a sign that the essential oil components were manufactured in the lab because enantioselective synthesis to make enantiopure compounds is more complex than synthesis of racemes. Enantiopure synthesis makes the adulterants less affordable, and not a viable cost cutting alternative [5]. However, natural enantioenriched and/or racemes do exist, such as the dihydrotagetone dominated essential oils of *Phebalium verrucosum* I.Telford and J.J.Bruhl [73].

## 4. Chemical Analysis of Essential Oils

Modern technology for studying essential oils has made chemical identification more accurate. Before the 1950s, natural product chemists had to work extremely hard to elucidate the structure of volatile compounds. Consequently, there were significant inconsistencies in the literature in chemical assignments of natural products. In modern times these have been rectified, for example, it was eventually demonstrated that (−)-ngaione and (+)-ipomeamarone (Figure 11) were merely enantiomers [18,74]. There was also considerable overlap in the naming of sesquiterpenes from South African species of *Widdringtonia*, i.e., ‘widdrene’ was corrected to thujopsene, ‘widdrenal’ to thujopsenal, and ‘widdrenic acid’ to hinokiic acid, leaving only ‘widdrol’ (Figure 11) as an exclusively South African sesquiterpene [27]. 

There was also a disagreement over the actual identity of maaliol, which was disputed by many chemists who believed it was synonymous with ledol (Figure 11), however, this was eventually disproven [55]. Furthermore, because of its similarity to ledol, the antimicrobial sesquiterpene prostantherol [75,76] was not named until the year 1994 [77], despite being detected numerous times in essential oils from species in the genus *Prostanthera*. 

### 4.1. Gas Chromatography

Volatile organic compounds are analysed using gas chromatography (GC) coupled to any one of a variety of detection systems. Gas chromatography is the primary step in essential oil analysis because the mixture is separated into individual components. This technique uses a ‘column’ to achieve this separation. A small amount of the essential oil is injected at the start of the column and a gas pushes the mixture through the column to the other side where each of the separated components meet a detector, usually a mass detector (MS) or a flame ionisation detector (FID). 

As components move through the column, the column is slowly heated from a low starting temperature, typically in the range of 40–60 °C, then it is raised at a rate of 3–5 °C per minute until reaching a maximum temperature in the margin of 280–300 °C. An inert gas, such as nitrogen or helium, is pushed through the column, and essential oil components travel as vapours in the direction of gas flow. It is a common misunderstanding that essential oil components travel as gases through the column, probably due to the name of the technique ‘gas chromatography’, but the etymology of the name is related to the use of nitrogen or helium gas and does not describe the physical state of the essential oil components as they pass through the column. 

Because essential oil components exist as liquids/vapours in the column, they usually elute before reaching their boiling point, for example, the boiling point of spathulenol at atmospheric pressure is 297 °C. According to thermodynamic theory, in a pressurised column the boiling point of spathulenol is much higher, yet it usually elutes between 150 and 200 °C. Furthermore, the boiling point of limonene is 176 °C, but it usually elutes between 105 and 115 °C. Much larger components, such as volatile diterpenes and coumarins, have boiling points that are much higher than the operating conditions of the machine, i.e., incensole acetate has a boiling point of 420 °C, yet it is still eluted like other components in gas chromatography, but it requires a longer retention time. 

The physical state of essential oil components (and volatile derivatives) as they travel through the column is determined by the inlet gas pressure (helium or nitrogen), the intermolecular interactions with the stationary phase (the column), temperature, and the concentration and vapour pressure of each individual molecule [78]. When essential oil components are travelling through the column, they are in a vapour phase, but when they are stationary, they are adsorbed into the column matrix, as a liquid. 

Most essential oil components change into vapour when the column temperature is raised high enough to break the intermolecular interaction with the column stationary phase. Because there is a very small quantity of the molecule in the column, when that temperature is reached, the whole liquid becomes a vapour. Thereafter the vapour exits the column and meets the detector. 

Figure 15 is an example of a gas chromatogram, which depicts peaks that formed along an axis of retention times (*x* axis). The retention time corresponds to the exact moment that the component exited the column and entered the detector. 

### 4.2. Gas Chromatography Stationary Phases (Columns)

Separation of essential oil components in chromatography may be additionally influenced by the packing inside of the column (stationary phase), which can be interactive (polar or chiral), or non-interactive (non-polar). The column is generally a thin hollow tube that has the appearance of a wire, typically 30 metres long and coiled into a circle for convenience and placed in an ‘oven’ which is a temperature-controlled cabinet. The hollow space of the column is packed with either a polar or non-polar stationary phase; either a polyethylene glycol (wax) or phenyl/methyl polysiloxane base, respectively. A list of commercially available columns and their respective polarity is given in Table 1. 

Out of the non-polar columns the numerical value in commercial names, such as ZB-1 and HP-5, gives an indication of the relative amounts of phenyl relative to methyl polysiloxane, where ‘-1 is a dimethyl polysiloxane and ‘-5 is a diphenyl dimethyl polysiloxane. Generally, as the numbers increase the interactivity (polarity) increases, but the changes are so subtle that the stationary phase is still regarded as non-polar. Nevertheless, this can create slight differences in retention time, but major differences will be evident by using a polar column and so this must be taken into consideration when judging the identities of eluting peaks according to retention times or arithmetic indices. 

### 4.3. Mass Spectrometric Identification by Comparing to a Mass Spectral Library

Essential oil components are identified according to two primary diagnostic methods. The first is the ‘fingerprint’ or pattern of fragments produced by the mass spectrometer (Figure 16). When a mass spectrometer is coupled to a gas chromatographer it is called gas chromatography–mass spectrometry (GC-MS). 

In gas chromatography, the mass spectrometer uses a beam of electrons to ionise the molecules in a technique called electron impact ionisation. Hence, the compound is bombarded with a carefully measured load of electrons, typically 70 millivolts, and the masses of the fragments are then detected. In Figure 16 the mass spectral pattern also includes a molecular ion peak (M^+^) which is the intact molecule at 210 Da. This fragmentation pattern is then compared against a library of spectra to detect a match. The mass fragmentation pattern in Figure 16 was not identified against the commercial NIST library because it is derived from a rare compound. It was isolated and identified by nuclear magnetic resonance spectroscopy such as 1-acetoxymyodesert-3-ene, a rare monoterpene iridoid (Figure 17) [18]. 

As a rule, diastereomers can be separated in gas chromatography, but not enantiomers, unless a chiral column is used. It is a common mistake for researchers to publish the identity of a component with its chiral descriptor, such as *d*-limonene or *d*-camphor. However, this is not possible, even if the database suggested it; these components can only be reported as limonene and camphor, respectively, unless they were isolated and determined in polarimetry. 

### 4.4. What Name to Use from the NIST Search Results

For those who have recently started analysing natural volatiles and essential oils by GC-MS, interpreting the names provided from the NIST library database can be confusing and at times, apparently contradictory. However, there are some simple logical steps that can be followed to aid in the decision-making process, when deciding what name to use from the list of names put forth from the NIST library. 

In the first instance, the NIST library provides a suggested compound identity, thereafter it is up to the scientist, or student, to decide if the match is correct, based on complementary evidence. With studies of essential oils, it is imperative to take into consideration the confidence provided by the software, where >90% confidence is regarded as a comfortable level to proceed to the next steps in verification. There are uncommon examples where >80% is also an acceptable albeit low level of confidence, such as when the metabolite is at a very low relative abundance, which can cause the background ‘noise’ of the detector to interfere slightly with the reproducibility of the mass spectral pattern. Another cause of low confidence is if the mass spectral instrument has not been calibrated recently. Consequently, mass spectral data loses precision, because the voltage of ionisation (electron impact) is not equivalent to the command that is put into the method, such as −70 mV. 

After the confidence of the suggested match from the NIST library is validated, the next important step is to calculate the arithmetic index (syn. retention index or Kovats index) and to compare it to a known value. The details of arithmetic indices are provided in the next section, however, if the column used is non-polar, such as a HP-5MS or DB-5 or something along those lines (see Table 2), then the published values can be taken from the seminal work provided by Adams [79]. A comparison to the published library from Adams is sufficient to qualify that the compound is highly probable, where the arithmetic index value is within 10 units of the value published by Adams. On rare occasions it can be a difference of 20 units, or even 30 units, provided that the difference is consistent with other compounds detected on the same chromatogram. However, such vast differences only occur when significantly different temperature programs are used, i.e., a temperature ramp of 20 °C per min, compared to a ramp of 5 °C per min, can make the arithmetic indices significantly different.

After it is determined that a name set forth by NIST is the most likely candidate identity, it is necessary for the researcher, or student, to compile the information into a coherent report, and this is where the confusion about which name to provide presents itself. There are three types of names that appear in the list of candidate identities. 

(1)Common name, which is the old convention where the person who first discovered the molecule typically named it with etymology related to the species from where it was isolated. For example, pinene was isolated from an essential oil produced from a member of the genus *Pinus*, and the ‘-ene’ in the name represents a double bond in the molecule.a.Common names are habitually provided with a stereochemical descriptor, i.e., α-pinene, γ-eudesmol, or δ-cadinene. As previously mentioned, these ‘isomers’ are usually a consequence of the position of a double bond, but on occasion they can also represent epimers, or an isomer that is not an enantiomer. b.These achiral isomers have slightly different mass spectral fragmentation patterns (signatures) and they also elute with different arithmetic indices, so they are usually validated by triangulation of these two metrics (arithmetic index and NIST match) with retention times of confirmed compounds eluting nearby on the chromatogram (within close retention times). c.A common name is usually short, and has no numbers in it, just English letters that are often combined with the Greek alphabet.(2)Common name hybridised with IUPAC nomenclature, which is a larger descriptor that is built on top of a common name.a.For example, germacrene D-4-ol, terpinen-4-ol, or methoxymyodesert-3-ene [18,80]. These are molecules that are very similar to another molecule that received a common name but are distinguished by a hydrogen deficiency (a double bond) or oxidation (an alcohol group or ketone, etc.). A number is used to specify the carbon number in the molecule where the difference occurs, relative to the namesake compound.b.Numeration of the derivatives of common name compounds does not follow IUPAC rules, it remains consistent with the original common name, i.e., methoxymyodesert-3-ene and myodesert-1-ene are numerated according to myodesertene, even though the numeration changes according to IUPAC rules.(3)Common name or hybrid common name that is given an enantiomeric descriptor.a.When the NIST library was created, authentic reference standards were used to build the database of mass spectral fragmentation patterns. b.It was common for authentic standards to be identified to the exact enantiomer, so when the reference was recorded the enantiomeric descriptors were included in the name, which was saved to the database. c.The enantiomeric descriptors, such as the plus sign ‘+’ or the R or S descriptors, are an artefact of the data entry process, when the data was being compiled into a library of mass spectral data (such as the NIST library). d.Enantiomeric descriptors, such as (+)-α-pinene, also known as 1R,5R-α-pinene, are determined using analytic methods, such as polarimetry or chiral GC, but are not determined using routine GC-MS. e.Thus, when a NIST match suggests a molecule with an enantiomeric descriptor, such as (+)-α-pinene, also known as 1R,5R-α-pinene, it is important to delete the descriptor and claim only the common name, or the hybrid common name, i.e., α-pinene, not (+)-α-pinene, or 1R,5R-α-pinene. This is because it is impossible to be this specific using only GC-MS.f.From a NIST library match, delete R or S, D or L, *d*- or *l*-*,* + or −, and simplify to the common or IUPAC (or systematic) nomenclature. (4)IUPAC nomenclature (also known as systematic nomenclature) is a convention established to standardise chemical names for an international audience. The long name is ‘International Union of Pure and Applied Chemistry’. The IUPAC nomenclature was established when common names became too frequent, causing some names to be used twice to describe different things.a.For example, the common name brevifolin can mean two different molecules, either brevifolin (geranium) which is 7,8,9-trihydroxy-1,2-dihydrocyclopenta[c]isochromene-3,5-dione in IUPAC nomenclature, or brevifolin carboxylic acid, which is 1-(2-Hydroxy-4,6-dimethoxyphenyl)ethan-1-one in IUPAC nomenclature. b.In IUPAC nomenclature, α-pinene is called 2,6,6-Trimethylbicyclo[3.1.1]hept-2-ene, and with an enantiomeric descriptor it is (1S)-, or (1R)-2,6,6-Trimethylbicyclo[3.1.1]hept-2-ene.c.Evidently the enantiomeric descriptor needs to be removed, i.e., delete (1S)-, or (1R)- and keep the remainder of the name, unless other work was done to confirm the chiral identity.

The NIST suggestion is often a common name, most likely with an enantiomeric descriptor, a modified common name, or a long and tedious IUPAC name. Conventionally, in report writing we avoid using the long and tedious IUPAC names and prefer the common names or their derivatives. When the NIST library gives us the IUPAC name, we are oblivious to the common name, which can be stressful. However, to know of the common name one can visit the ‘NIST Chemistry Webbook’ [81], which is an online database that provides gas chromatography data, mass spectral data, and synonyms of the IUPAC name. From the list of synonyms, a common name can be chosen, preferably one without enantiomeric descriptors.

### 4.5. Calculation of Arithmetic Indices

The second diagnostic method used in gas chromatography is arithmetic index (AI) also known as retention index. It is necessary because the NIST library might suggest several very convincing matches by comparing mass spectra. Figure 18 is an example of two different mirror matches, which demonstrates the similarity between mass spectral information of two components that are like globulol but are not globulol. This confirms the importance of another tool for compound identification.

The next discriminating tool is retention time. However, because retention time will be different between the different instruments and methods used, the retention time is standardised by creating a relative retention time calculation, or index. This is calculated with the help of a homologous series of n-alkanes (Figure 19, C15 not included). The series of n-alkanes are injected onto the GC-MS column using the exact same operating conditions as for the essential oil. This makes them comparable. The equation for AI determination is given below.
$$AI=100\times \left(a+\frac{RTz-RTa}{RTb-RTa}\;\right)\;$$

*RT* is retention time, i.e., *RTa* is retention time of n-alkane eluting before *z,**z* is the essential oil component,*a* is the carbon number of the n-alkane eluting before *z*,*b* is the carbon number of the n-alkane eluting after *z*.

The AI is calculated for every component in an essential oil. The retention time of each component is between the retention time of two n-alkanes, i.e., limonene elutes after decane (C10) but before undecane (C11), so limonene elutes between C10 and C11. If limonene is denoted as z in the formula above, and a = 10, b = 11, then the retention times (RT) are as follows, according to Figure 20, RTa = 8.08 min, and RTb = 10.74 min. If the retention time of limonene is 8.93 min, then the calculations are as follows, 100 × (10 + (8.93 − 8.08)/(10.74 − 8.08)) = 100 × (10 + 0.32) = 1032. If the GC-MS has a non-polar column installed, this value can be compared to the value given in the book by Adams [79], which is 1024 for limonene. Because this value is within 10 points it may be considered a tentative match. Alternatively, values for both polar and non-polar columns can be found on the NIST Chemistry WebBook website [81].

It is a daunting task to calculate arithmetic indices for all components in a mixture, particularly if multiple essential oils are injected. However, a useful calculation on an excel spreadsheet can make the process fast and simple (Figure 20). The excel formula for each of the cells in Figure 20, is given in Table 2. These formulas enable automatic arithmetic index calculation. Because modern GC-MS operating software can export a spreadsheet of retention times and suggested NIST library matches, with probability of the match, then the retention times can be cut and pasted into this excel spreadsheet in Figure 20, and the cells copied an infinite number of times down the page to cater to hundreds of retention times in a single step.

### 4.6. Other Techniques in Chromatographic Analysis

Quantitation of essential oil components in GC-MS is not regarded as absolute, it is merely for qualitative comparison. Mass spectral data is not useful for accurate quantitation because it is not comparable between the different types of compounds [82]. Salient inconsistencies are evident between monoterpenes and sesquiterpenes, or terpenes and phenylpropanoids when GC-MS quantitation is compared to actual values by quantitative NMR or by using authentic reference standards.

The major challenge to quantitative analysis from GC-MS is the inaccuracy of the mass spectral data itself. To resolve this, authentic standards are used at known concentrations to quantify absolutely or semi-quantitively, which are injected using the same method as the essential oil. Furthermore, flame ionisation detection (FID) is regarded as more accurate in quantitative measurement. In this technique, the essential oil component that is eluting from the column is directed into an ignited hydrogen gas flame. The intensity of the flame is measured using two electrodes that detect the ion cloud, which gives the value used to represent the chromatographic intensity. Generally, GC-FID is only used when the identity of the components is known, either for quantitative comparison between species or chemotypes, and to obtain more accurate information of the accurate quantity of components.

A less sophisticated technique that is often used in perfumery is olfactometry. This method of detection uses the human nose in a process called GC-olfactometry [83]. It creates a subjective record associated with the odour of individual components. Olfactometry is also commonly used with a chromatographic technique called ‘two-dimensional gas chromatography’ [83], which connects two columns with different stationary phases, i.e., polar, and non-polar. This helps when components are not separating well. Nevertheless, while there are many methods used in essential oil analysis, GC-MS is by far the more common tool.

## 5. Authentication of Essential Oils

Counterfeiting and adulteration of essential oils is dishonest, a breach of ethics, and may be unsafe to the consumer [84]. However, if consumers become aware of the tell-tale signs of dishonest marketing, the demand for counterfeited items reduces and the incentive to counterfeit is reduced [5,85].

### 5.1. Analytical Methods Used for Authentication of Essential Oils and Natural Volatiles

In the authentication of essential oils, it is common for mere chemical characterisation to be performed. However, adulterators have learned to defeat these ‘first pass’ chemical checks by crafting essential oils that are prima facie chemically identical, or by using closely related species to produce essential oils. It is sometimes possible to detect when there is counterfeiting by examination of the enantiomeric ratios of components [86]. With reasonable variation the enantiomeric ratios of components are consistent for the naturally occurring ‘enantiotypes’ in species.

The type of column used for enantiomeric analysis is a chiral column. Chiral chromatography uses a chiral-selective cyclodextrin incorporated into the stationary phase, which interacts differently with enantiomers. One of the applications is to help differentiate two similar essential oils according to their ‘enantiotype’. For example, this was done to distinguish between two frankincense essential oils, one from *Boswellia carteri* Birdw and the other from *Boswellia sacra* Roxb ex Colebr. They were chemically very similar, but the α-pinene of *B. carteri* was almost a raceme (equal mixture of both enantiomers) whilst that of *B. sacra* was overwhelmingly dominated by the positive enantiomer [87]. As previously mentioned, a study of enantiotypes can reveal if the essential oil was adulterated with a synthetic version of a component, because cheap synthetics are often racemes that interfere with the natural enantiomeric ratios defined in earlier studies [86]. Researchers should also be aware of naturally occurring enantiotypes in essential oils, such as the enantioenriched and enantiopure types of *Phebalium verrucosum* I.Telford and J.J.Bruhl [73,88].

A summary of the enantiomeric ratios of citrus essential oils has been provided by Do et al. [89]. If the requirement for enantiomeric purity is that one enantiomer is >94% of the other, then by comparison with α-pinene, the components β-pinene and limonene are more often enantiopure. In citrus essential oils, β-pinene is enantiopure in six out of nine samples, while limonene is enantiopure in 9 out of 11. This is in contrast with natural α-pinene, which is more commonly enantioenriched across species of citrus, with only 4 out of 11 specimens demonstrating enantiopurity.

Another chemical test that can help to determine if essential oils are adulterated is carbon isotope ratios. Where isotopic analysis is used to authenticate an essential oil, a specialised mass spectrometer is required. The type of column used for isotope ratio analysis is generic, however, the MS detector is not; it is dedicated to isotopic analysis and is denoted isotope ratio mass spectrometry (IRMS). There are two factors influencing isotopic ratios, with one being the addition of synthetic components that were made from fossil fuel-derived scaffolds. In this case the natural process of radioactive decay substantially reduces the amount of ^13^C in fossil fuels over time, which reduces the ^13^C/^12^C ratio. However, it is more common for IRMS to be applied where multiple essential oils of natural origin are combined to counterfeit a more expensive essential oil. In this case, the difference in isotopic ratios is a consequence of the different classes of photosynthesis utilised by the plant. Generally, CO_2_ capture during photosynthesis is biased toward heavier carbon atoms, i.e., ^13^C in preference to ^12^C, however, plants that follow C3 photosynthesis (Calvin cycle) do not accumulate as much ^13^C as C4 plants (Hatch–Slack cycle). Hence, when lemon essential oil from a C3 plant (*Citrus x limon*) is adulterated with citral from a C4 grass species (*Cymbopogon citratus* (DC.) Stapf) the isotopic ratio is biased toward higher ^13^C [90]. A third type of photosynthesis follows the Crassulacean cycle, which utilises both the Calvin and Hatch–Slack cycles, creating an isotopic ratio in between that of C3 and C4 plants. Hence, for IRMS to be truly effective the technical knowledge of the authenticator must be well developed.

Other techniques that have proven useful in the authentication of essential oils include spectroscopic techniques, such as middle infrared spectroscopy [91], nuclear magnetic resonance spectroscopy [92], and even UV-fluorescence coupled to transmittance, which was used for the authentication of patchouli oil [93]. However, spectroscopic methods will not provide evidence prima facie, to dispute sophisticated adulteration.

### 5.2. Simplistic Methods for Authentication of Essential Oils

A comprehensive review published in 2015 provided a significant summary of adulteration tactics that commonly occur in the industry for essential oils [89]. What is conveyed is that there is no detailed protocol to follow that is generic for all the kinds of adulteration. Each of the natural products that can be adulterated have tell-tale signs that are determined on a case-by-case basis, and the analytical methods are also context specific. For example, adulteration with synthetic linalool will contaminate the essential oil with artefacts of the synthetic process, which are dihydro- and dehydrolinalool.

There are several non-sophisticated methods for the authentication of essential oils, where the vendors or merchants have not used a ‘sophistication’ to hide their attempts to cheapen their products [5]. In such cases, only simplistic methods are necessary to determine if the product is a counterfeit item. Although simplistic, basic laboratory equipment is necessary in most cases (except in evaporation tests). Much information can be harnessed from infrared determination, ultraviolet absorbance (UV spectrophotometry), or thin layer chromatography. Because these items are the most basic laboratory techniques, they are considered non-sophisticated.

#### 5.2.1. UV Absorbance Determination Using Spectrophotometry

In another example, dilution of citrus pressed oils with cottonseed oil will dilute the UV active components, the coumarins. The concentration difference can be determined by UV spectrophotometry. Similarly, when any essential oil containing phenylpropanoids is diluted with a carrier oil, the concentration difference can be measured using a UV spectrophotometer. Any component with delocalised electrons will have a UV maximum (peak absorbing wavelength, λmax) that can be utilised. Essential oil components that have a benzene moiety can be examined in this way, such as the terpenoid components, *p*-cymene, thymol, carvacrol, cumin aldehyde, or α-turmerone. The UV absorption value should be many orders of magnitude different compared to the natural authentic equivalent essential oil, because small differences can be related to natural variation in relative abundance of components.

Common phenylpropanoids include anethole, eugenol, safrole, elemicin, or cinnamaldehyde and all of these have UV chromophores that absorb at wavelengths (i.e., eugenol = 280.9 nm) that are similar by comparison with most of the aromatic terpenes (i.e., thymol = 275 nm). Alternatively, when essential oils are not expected to have a UV chromophore, the spectrophotometer can be used to determine if the oil includes fixed components, such as flavonoids or coumarins, which will indicate that the product is not an essential oil but an absolute or pressed oil.

#### 5.2.2. Evaporation Ability

Another way to determine if a product that claims to be an essential oil is not an essential oil is by testing for evaporation ability. For example, a carrier oil combined with an essential oil, a pressed oil, or an absolute, will not completely evaporate. This is because there will be non-volatile components if the product is not an essential oil, so by placing a few drops of the product onto paper and heating it with a hair drier for 5 min, the authenticity can be revealed. If some of the oil remains on the paper, then some of it is not volatile, so it is either not an essential oil, or it is adulterated. However, according to the international standards organisation, if a pressed oil is produced from citrus rinds, such as bergamot, it is called an essential oil, even if it was not hydrodistilled. Thus, if bergamot turns out to be a pressed oil, it is still authentic, just not hydrodistilled.

#### 5.2.3. Thin Layer Chromatography

As an alternative to spectrophotometry, thin layer chromatography can be used (TLC) can be used to authenticate essential oils. Crude extracts often contain components that are more polar than essential oil components, so TLC can be used to reveal these. Any component that stays at the baseline of a TLC plate using 10% ethyl acetate in 90% hexane, is too polar to be an essential oil component, and must, therefore, be from extraction, not hydrodistillation. If the TLC mobile phase is changed to 20% ethyl acetate, 80% hexane, then components below an Rf value of 0.4 are impossible in an essential oil.

However, common authentication techniques will also detect components that are either volatile, or relatively apolar but with a vapour pressure that is too low to be volatile. In such cases, the authenticator needs to know what to look for to detect such cases of adulteration. This relies on familiarity with the adulteration techniques that are used by counterfeiters [5]. For example, if the authenticator knows that lavender essential oil is often adulterated with grapefruit skin pressed oil, then TLC can be used to determine if coumarins are present. The coumarin auraptene can be determined by use of a UV indicator in the TLC plate (F254). Alternatively, ammonia gas can be fumigated over the normal silica gel plate, and this will make the component fluoresce under UV light. If the authenticator has a sample of auraptene it can be used as an authentic standard to compare, using TLC migration (Rf value), then the outcome is more credible.

## 6. Suggestions and Concluding Remarks

Because the field of essential oil research is inclusive of diverse science backgrounds, a comprehensive and concise introduction to the chemistry and analysis is timely. The student, or researcher, will benefit from an understanding of how to interpret and use data from analytical instruments, and how to differentiate between different types of molecules considering their molecular formula, structures, parent group, enantiomer or diastereomer, and so forth. Additionally, of importance is the empowerment of individuals to recognise and avoid counterfeited, adulterated, or misrepresented products in the marketplace. In this regard, authentication of essential oils and natural volatiles has become an industry of its own.

Currently the published literature that describes essential oils is populated by works that fail to present the data with calculated retention indices (syn. arithmetic index). This is the first indicator that the data has not been interpreted and corrected, but rather, reported without modification from the library search report. Authors need to be reminded that NIST library matches are merely suggestions that need to be confirmed or nullified by comparing calculated retention indices against published values. A valuable resource is provided by Adams for making this comparison [79]. Authors should also avoid using the enantiomeric descriptors (R, S, +, −, D, L) that are suggested in NIST library searches. These values need to be removed from the compound identities unless further work is done to determine their respective enantiomer identity.

In the published literature there is also some misunderstanding over the relative quantities of components in essential oil mixtures. The values determined by integration of a chromatogram, generated from mass spectral detection, are not absolute; they are merely qualitative and are used as a general guide on the quality of an essential oil. The values are also affected by the integration parameters used, which can be adjusted to select peaks above a specified quantitative value. Minor components that are not recognised by the integrator will not be included in the quantitative analysis. Thus, authors should avoid reporting their quantitative values to several decimal places. A single decimal place is used for convention, but in reality, the quantitative value is only useful as a whole integer.

Significant natural variation of essential oil component quantities is expected. Even if the same essential oil is analysed two times, the values of component quantities might change by several integers. More significant differences will be evident between two specimens of the same species. Even when plant material is harvested from the same plant specimen to make essential oil for a second time in succession, the respective essential oil will be different to the first. Thus, authentic essential oils do not need to adhere to exact quantities specified in the published literature.

Conversely, authentication scientists should be aware that confirming a single known component in an essential oil does not automatically qualify it as authentic. For example, it is impossible to count the number of essential oils that contain 1,8-cineole, limonene, α-pinene, and so forth. However, detection of uncommon or rare components that belong to the essential oil increases the likelihood of authenticity.

## Figures and Tables

**Figure 1 plants-11-00789-f001:**
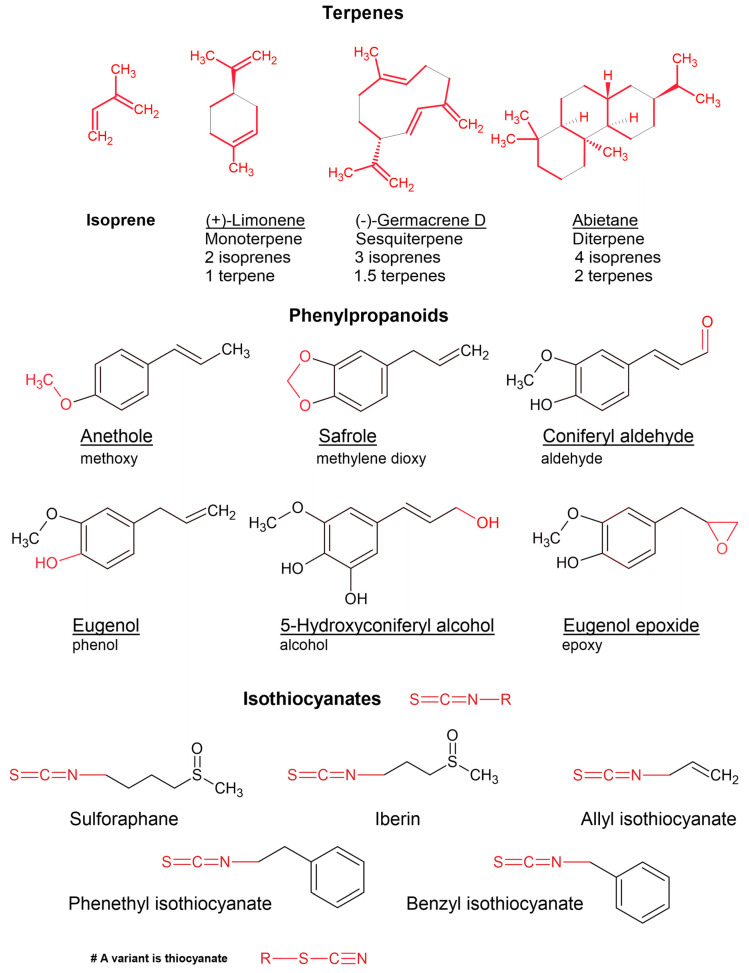
Examples of terpenes (with isoprene units indicated by red highlighting), phenylpropanoids, and isothiocyanates.

**Figure 2 plants-11-00789-f002:**
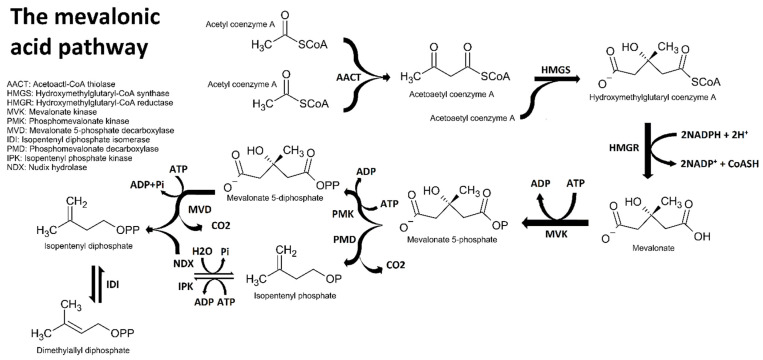
The biosynthetic steps of the mevalonic acid pathway toward the synthesis of terpenes.

**Figure 3 plants-11-00789-f003:**
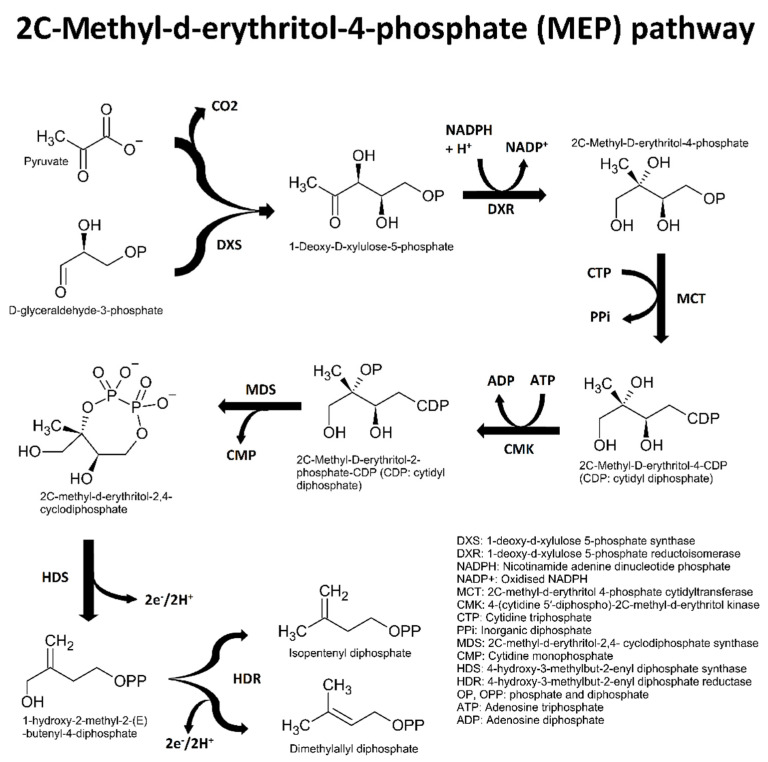
The MEP biosynthetic pathway that creates terpenes.

**Figure 4 plants-11-00789-f004:**
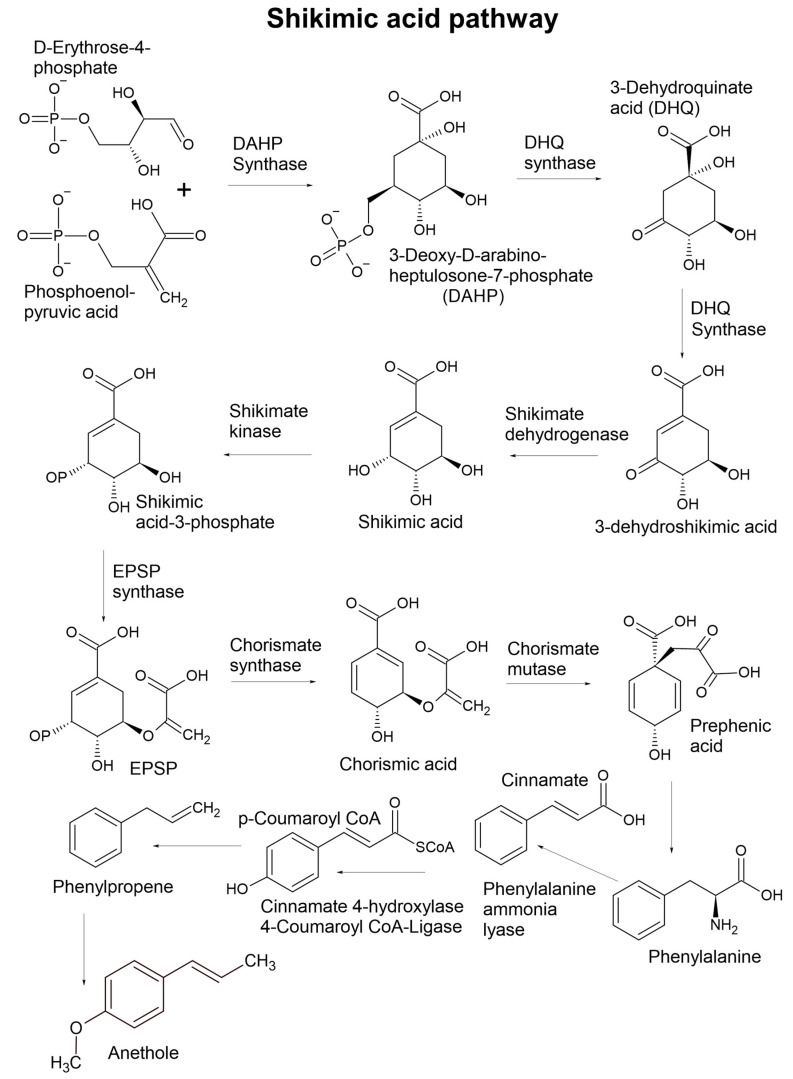
The shikimic acid biosynthetic pathway that creates phenylpropanoids.

**Figure 5 plants-11-00789-f005:**
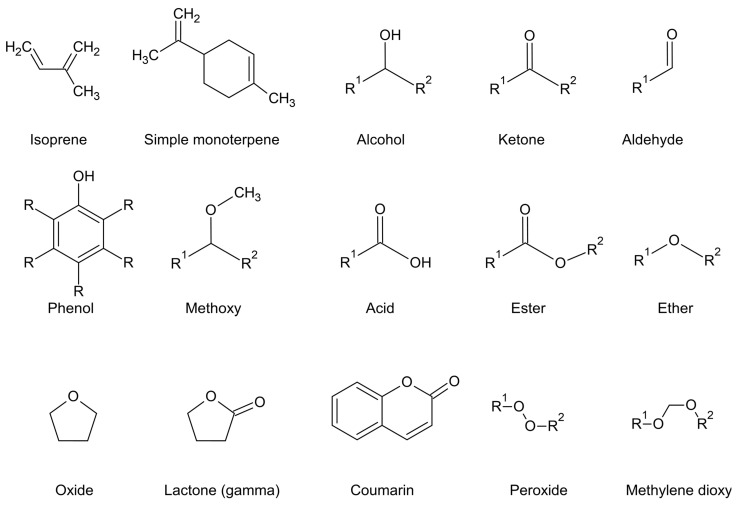
Character of oxidation and corresponding chemical class.

**Figure 6 plants-11-00789-f006:**
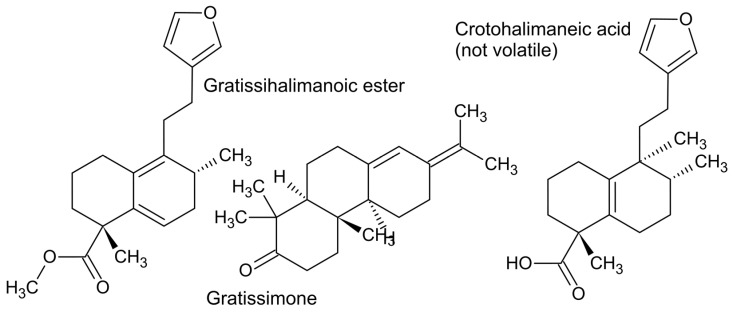
Two volatile diterpenes that comprise the major fraction of an essential oil from *Croton gratissimus*, sampled at the Pretoria Botanic Gardens, South Africa.

**Figure 7 plants-11-00789-f007:**
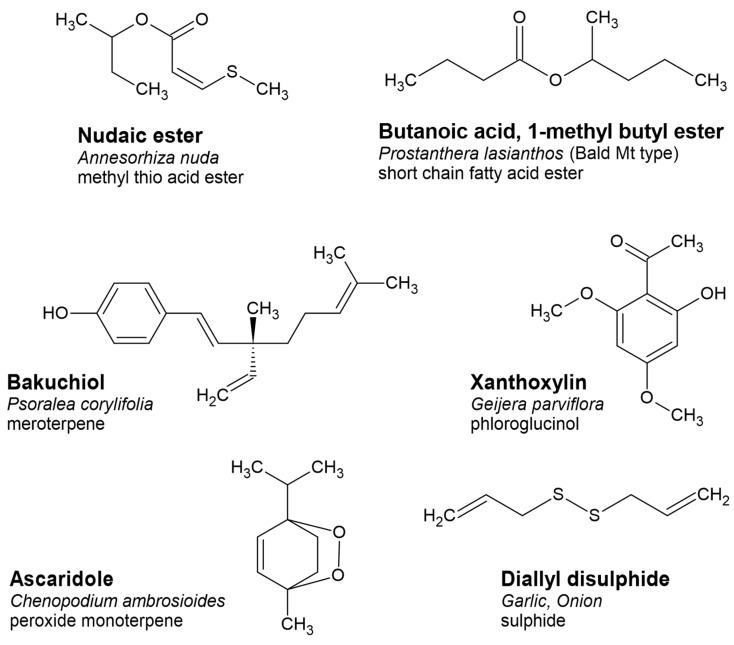
Unusual essential oil components.

**Figure 8 plants-11-00789-f008:**
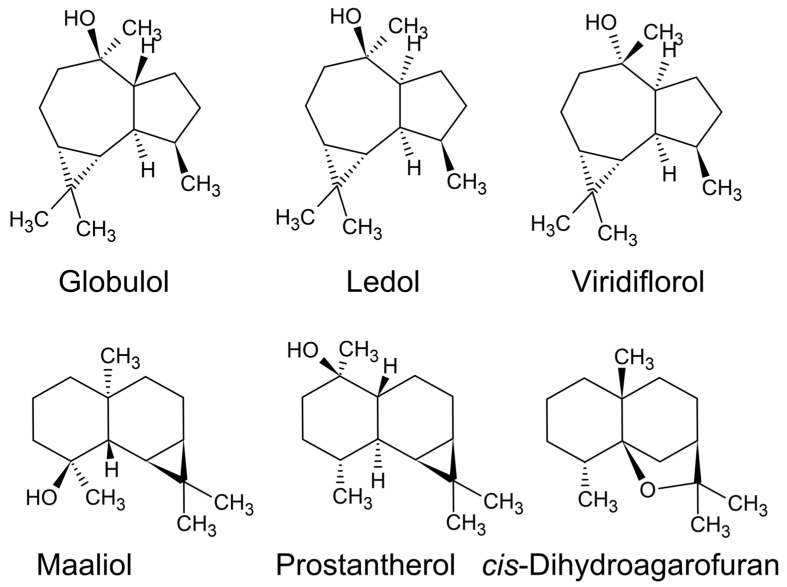
Sesquiterpene epimers and structurally related derivatives that can be difficult to differentiate by mass spectral patterns alone.

**Figure 9 plants-11-00789-f009:**
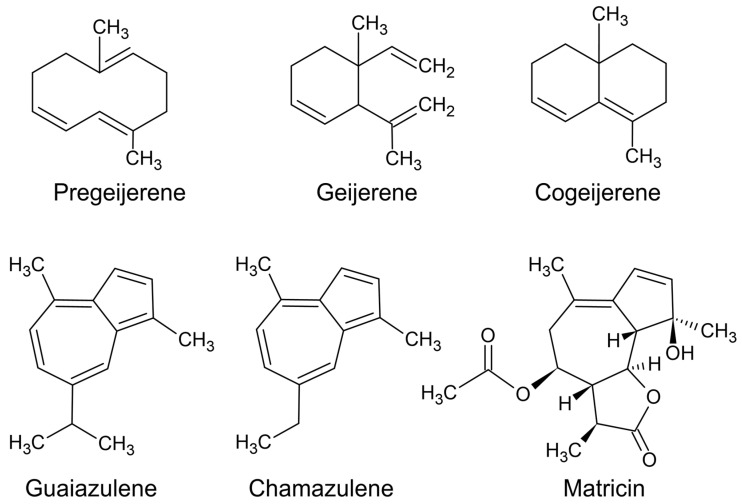
Compounds that confer colour to an essential oil. Green components are pregeijerene, geijerene, and cogeijerene; blue components are guaiazulene and chamazulene, and matricin is a heat labile precursor to chamazulene.

**Figure 10 plants-11-00789-f010:**
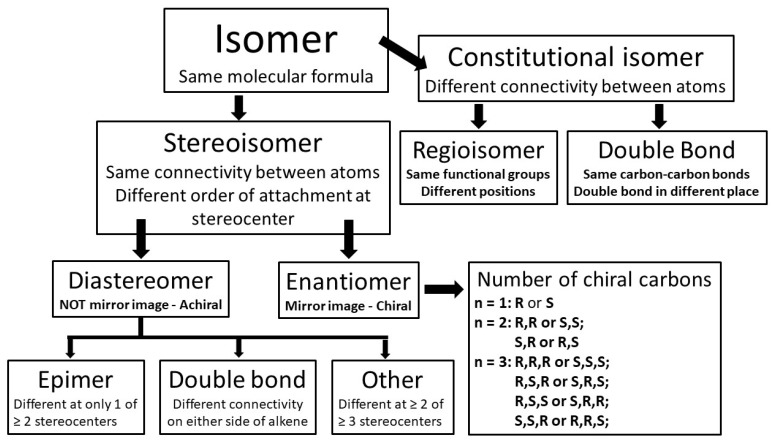
Isomer hierarchy in organic chemistry nomenclature.

**Figure 11 plants-11-00789-f011:**
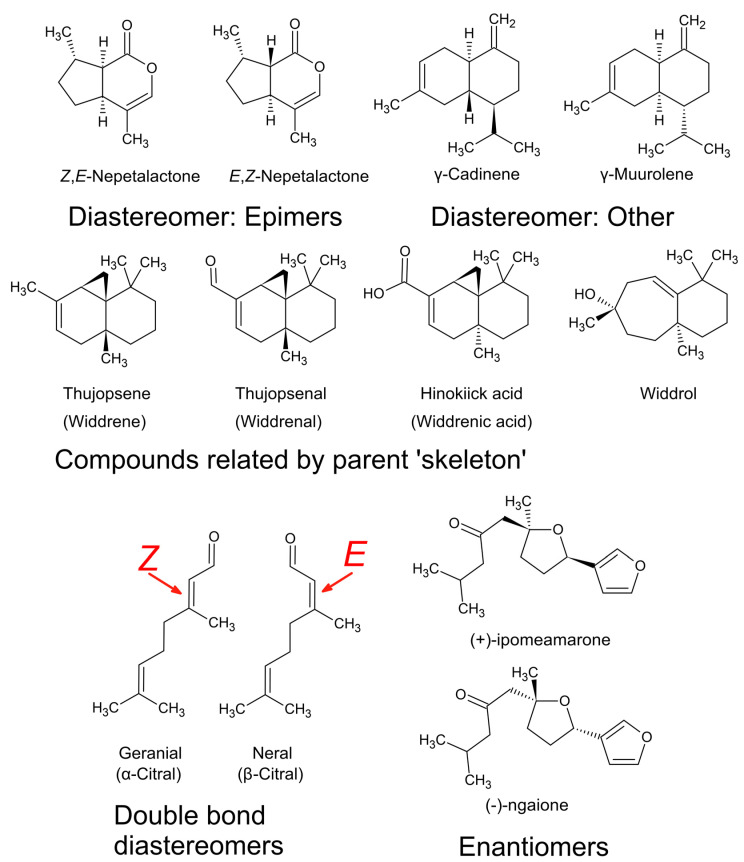
Examples of some diastereomers, enantiomers, and a series related by their parent skeleton, with their vernacular names.

**Figure 12 plants-11-00789-f012:**
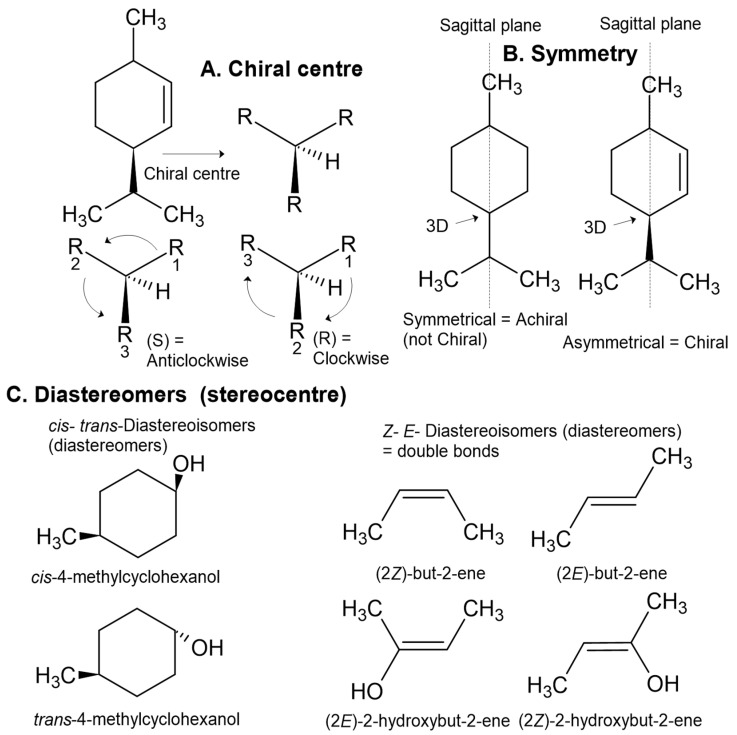
Chirality and cis/trans isomerism.

**Figure 13 plants-11-00789-f013:**
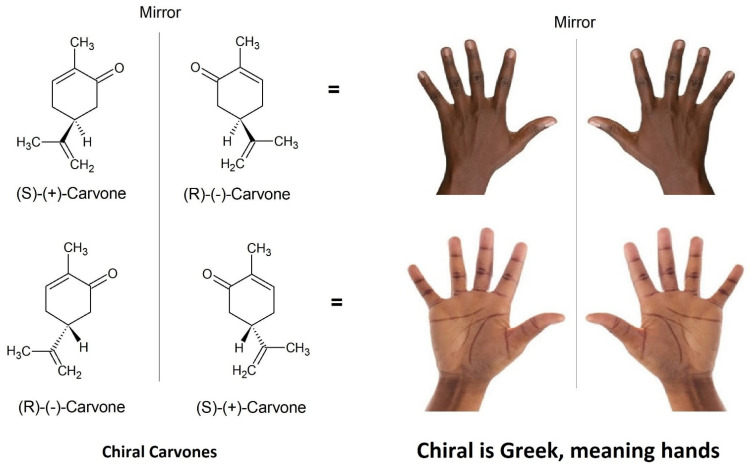
Chirality is also known as hands. Carvone is an example of a chiral compound that has only one chiral centre.

**Figure 14 plants-11-00789-f014:**
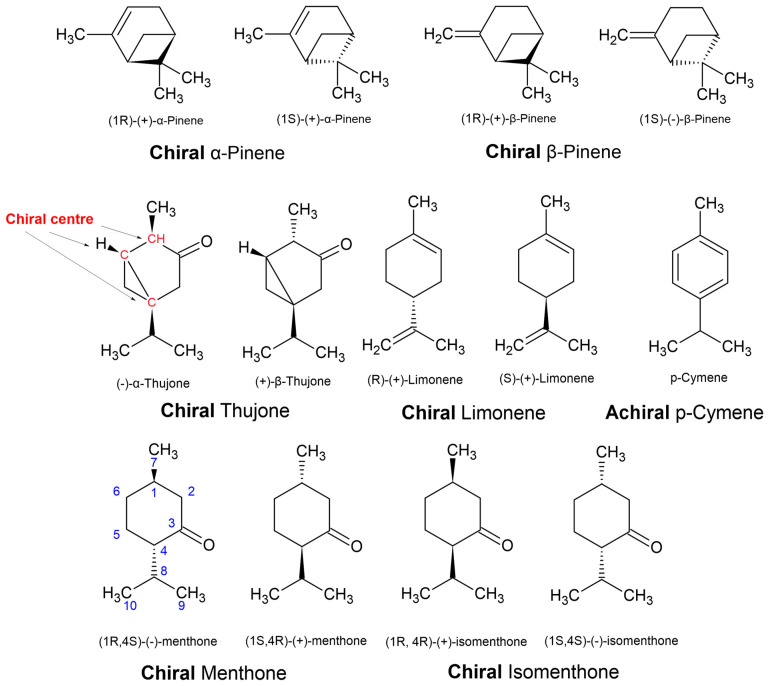
Enantiomers of chiral compounds, such as thujone, limonene, and menthone/isomenthone.

**Figure 15 plants-11-00789-f015:**
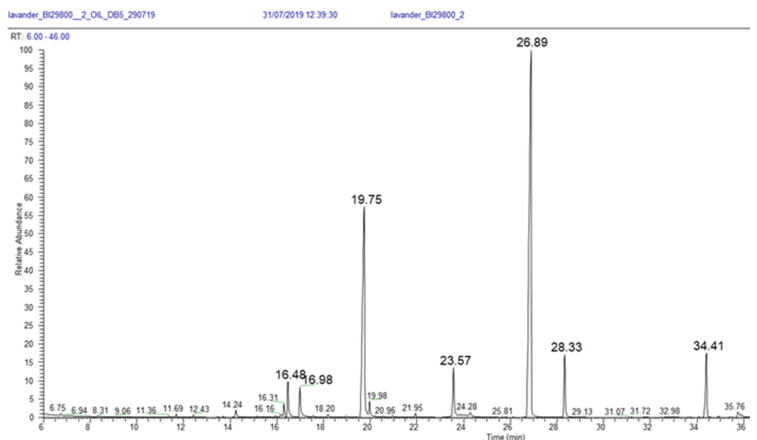
Gas chromatogram produced in gas chromatography, using a mass spectrometer for peak detection of a sample of lavender oil.

**Figure 16 plants-11-00789-f016:**
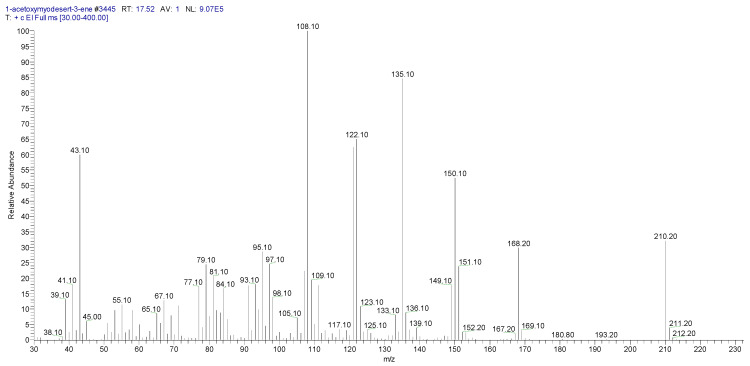
Mass spectrum of a rare compound, 1-acetoxymyodesert-3-ene, using electron impact ionisation.

**Figure 17 plants-11-00789-f017:**
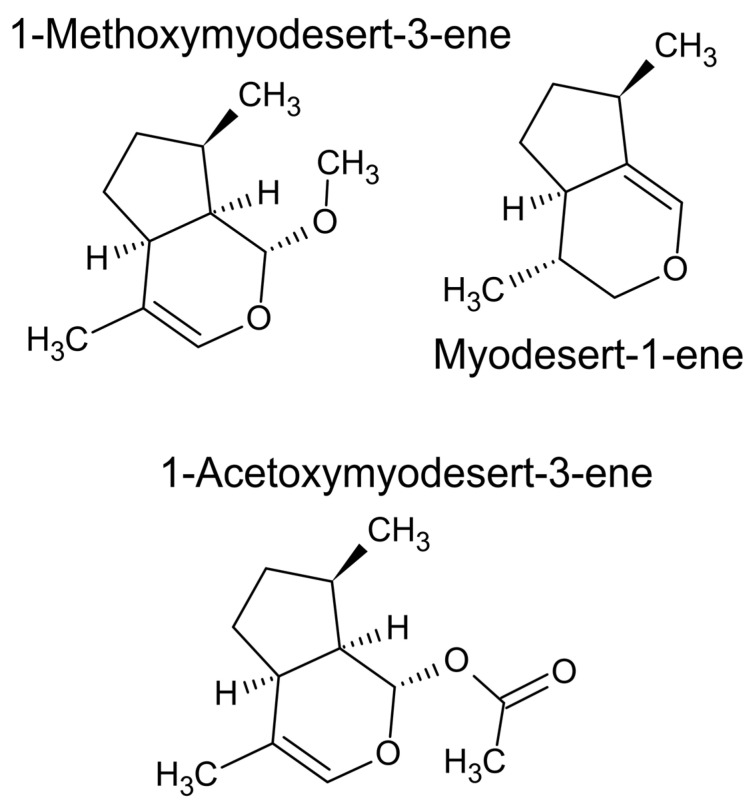
Unusual essential oil components from *Eremophila deserti*.

**Figure 18 plants-11-00789-f018:**
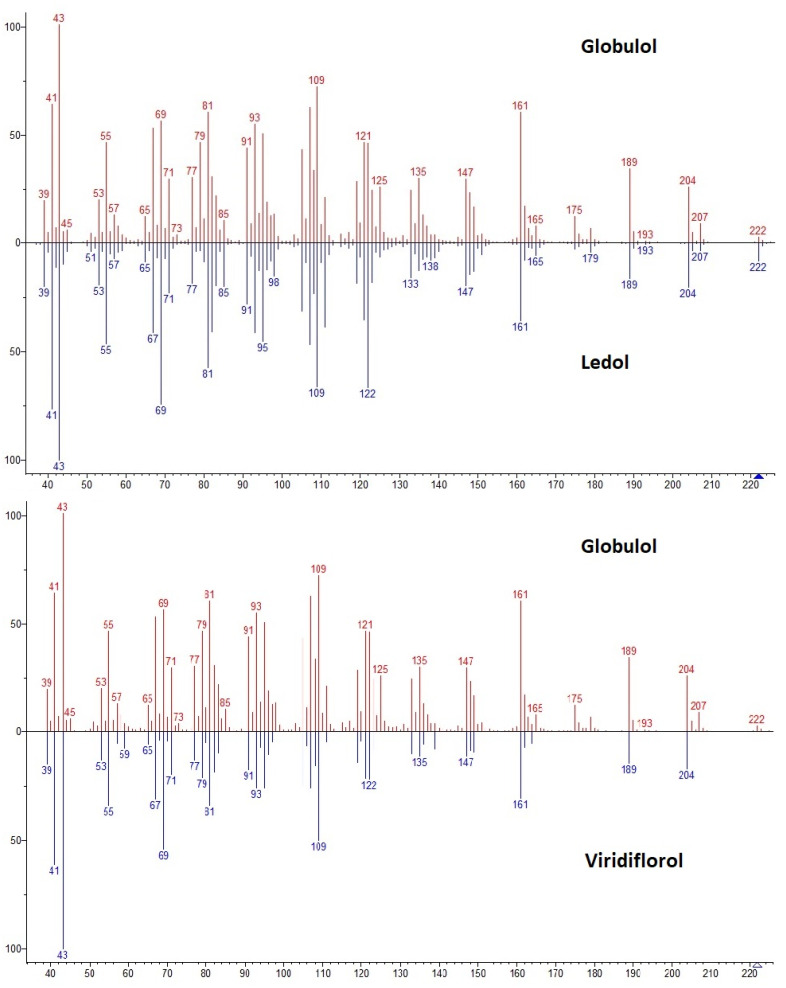
Examples of two different mirror matches between globulol and ledol (upper panel) and globulol and viridiflorol (lower panel).

**Figure 19 plants-11-00789-f019:**
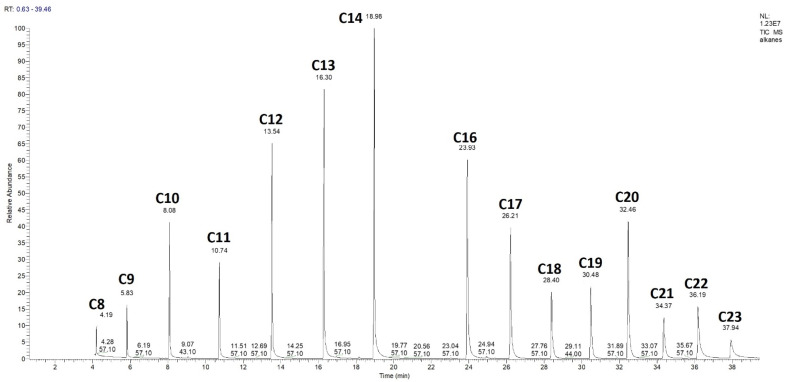
A series of n-alkanes used for calculation of arithmetic indices. This series goes from C8 to C23 but is missing C15.

**Figure 20 plants-11-00789-f020:**
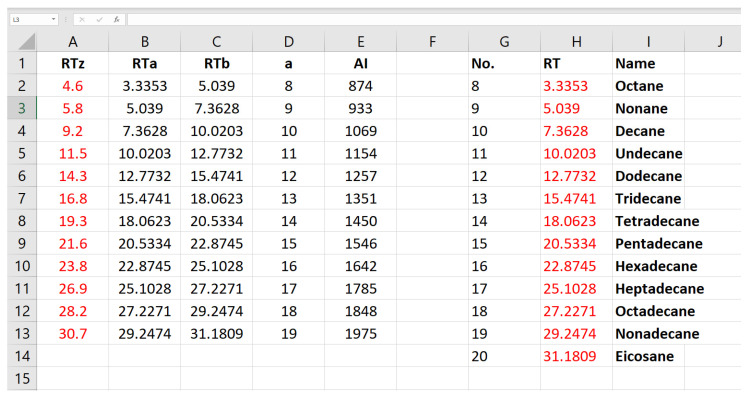
Excel spreadsheet for automatic calculation of arithmetic index by cutting and pasting of RT values (cells A2–A13). RTz is the retention time of the essential oil component, RTa is the retention time of the n-alkane that elutes before RTz, RTb is the retention time of the n-alkane that elutes after RTz, a is the carbon number of the n-alkane that elutes before RTz, AI is the calculated arithmetic index. Rows G–I are information of the retention times of the homologous series of n-alkanes.

**Table 1 plants-11-00789-t001:** List of gas chromatography column equivalents that are either non-polar or polar.

Non-Polar Columns	Polar Columns
ZB-1; DB-1; OV-1; SE-30; PB-1; OV-101; DB-5; DB-5MS; HP-5MS; BP-1; SPB-5; BPX-5; RTX-1	PEG-20M; PEG 4000; Carbowax 20M; Carbowax 4000; HP-Wax; DB-Wax; Supelcowax; Supelcowax-10; Innowax

**Table 2 plants-11-00789-t002:** Calculations used on the excel spreadsheet in Figure 20 for automatic calculation of arithmetic index.

Symbol	Cell ID	Excel Formula
RTz	A2–A13	Retention time value obtained experimentally from GC-MS chromatogram
RTa	B2–B13	=@IF(A2 > H$13,H$13,IF(A2 > H$12,H$12,IF(A2 > H$11,H$11,IF(A2 > H$10,H$10,IF(A2 > H$9,H$9,IF(A2 > H$8,H$8,IF(A2 > H$7,H$7,IF(A2 > H$6,H$6,IF(A2 > H$5,H$5,IF(A2 > H$4,H$4,IF(A2 > H$3,H$3,IF(A2 > H$2,H$2,error))))))))))))
RTb	C2–C13	=IF(B2 = H$2,H$3,IF(B2 = H$3,H$4,IF(B2 = H$4,H$5,IF(B2 = H$5,H$6,IF(B2 = H$6,H$7,IF(B2 = H$7,H$8,IF(B2 = H$8,H$9,IF(B2 = H$9,H$10,IF(B2 = H$10,H$11,IF(B2 = H$11,H$12,IF(B2 = H$12,H$13,IF(B2 = H$13,H$14))))))))))))
a	D2–D13	=IF(B3 = H$2,8,IF(B3 = H$3,9,IF(B3 = H$4,10,IF(B3 = H$5,11,IF(B3 = H$6,12,IF(B3 = H$7,13,IF(B3 = H$8,14,IF(B3 = H$9,15,IF(B3 = H$10,16,IF(B3 = H$11,17,IF(B3 = H$12,18,IF(B3 = H$13,19))))))))))))
AI	E2–E13	=100*(D4 + (A4-B4)/(C4-B4))
No.	G2–G13	Carbon number of alkane from homologous series of n-alkanes
RT	H2–H13	Retention time of alkane from homologous series of n-alkanes

RTz is the retention time of the essential oil component, RTa is the retention time of the n-alkane that elutes before RTz, RTb is the retention time of the n-alkane that elutes after RTz, a is the carbon number of the n-alkane that elutes before RTz, AI is the calculated arithmetic index. Rows G–I are information of the retention times of the homologous series of n-alkanes.

## Data Availability

Not applicable.

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
