# Peer review of "Fundamental Chemistry of Essential Oils and Volatile Organic Compounds, Methods of Analysis and Authentication"

_plants, 2022, doi:10.3390/plants11060789_

Round 1

Reviewer 1 Report

Paper is well written, presenting comprehensive info about analysis and authentication s of EOs.  Good Job! My minor remarks: 1. Add information about all enzymes, involved in presented biosynthetic pathway; 2. Correct IDP to IPP (I mean Isopentenyl pyrophosphate, Fig. 1); 3. Correct stereochemistry in Abietane (mark all or none chiral atoms); 4. Use one convention for chemical compounds e.g. for methyl group , I mean Me or "-"); 5. Add more precise info in with organism (organelles) ME pathway is performed; 6. Add all missed abberications for Fig. 2; 7. Correct Phosphoenolpyruvate structure; 8. Correct structure of DAP; 9. If authors mark stereoisomerism on chiral compounds, please do the same for e.g. Phenylalanine; 10. "Double bound", I mean mark of aromaticity in safrole, inside benzene ring; 11. Is numeration of 5-hydrocyconiferyl alcohol OK? 12. Correct structure of Allyl isothiocyanate; 13. Perhaps on Fig. 3 simple monoterpene? 14. Explain in manuscript "meroterpene"' 15. On Fig. 3, Chenopodium ambrosioides? 16. Explain "phloroglucinol". 17. On "Number of chiral atoms: Authors could mark also diastereoisimer pairs; 18. Add info, that nam "epimers" is mainly used for carbohydrates; 19. Correct to "Hinokiic acid"; 20. Correct stereochemistry for "Hinokiic acid" (methyl group); 21. Please provide other exaple for Chlorobut-2-ene (not natural compound) or butene; 22. Chiral "Carvones"?  (Fig. 8)?  23. Add latin ethymology for d and l? 24. Correct "D-(+)-Glucose" to " D-(+)-glucose" etc. L. 360; 25. For menthone, mark number of carbons (I mean which one is "1, 2,3,4,5,6 etc."; 26. " List of gas chromatography columns" or " List of gas chromatography columns equivalents"?; 28. Add structure for   "1-acetoxymyo- 517 desert-3-ene" as well as compounds on Fig. 12; 29. On fig. 13 I advice to remove main MS, and add number of C in hydrocarbons;

Author Response

Dear Review

Thanks for your very helpful revision recommendations. We have incorporated all of your corrections and given the manuscript I major revision to add more value. 

Reviewer 2 Report

Main remarks:
- in the introduction, it is necessary to clearly indicate the purpose and innovation of this work
- conclusions are too short, please elaborate
- all drawings and diagrams - please adjust the font style and size to be the same as in the text, also their proportions so that they constitute a coherent whole with the text
- references - please follow the Plants requirements carefully, it is good to have a doi number for each item
- please revise the language - typos appear

Author Response

Dear Review

Thank you for your time. 

The manuscript has been given a major revision, and the relevance of the work has been elaborated upon in the introduction and a longer conclusion has been written. 

Reviewer 3 Report

This manuscript presents a literature overview regarding the fundamental chemistry of essential oils and volatile organic compounds, methods of analysis and authentication. The authors should consider the below comments to improve their paper.

  1. Please insert a list of abbreviations which would be useful considering the notations used in the paper.
    2. The authors should reformulate the abstract in order to emphasize the novelty of the paper. I do not see what is the novelty and advancement in the field of essential oils. The literature overview is too fundamental to be useful for the readership of the journal. The paper does not present any novel applications regarding essential oils that are not already known ore could be a summary of what is already known. I recommend that the authors should consolidate the review with discussions and conclusions that could provide novel approaches to researchers working in this field.
  2. The section related to the use of the excel template seems to be under the performance level of the journal. It is something that can be presented and recommended for students but not for researchers working in this field. Please remove it or further improve it.
    4. Conclusion need to be revised and future perspectives should be defined and added.

Author Response

Dear Reviewer

Thank you for your examination of the manuscript. 

Some of the points raised have been addressed. We have improved the conclusion by expansion and we have reworded the abstract to make clear the novelty of this piece. 

The paper was written specifically for young researchers from developing nations. Most of the authors are from nations that are resource deficient. Hence, the fundamentals level is intentional, to be a resource to those people. 

The review aims to guide young researchers away from common errors that keep appearing in the literature. The language style is approachable and clear, to help with those who speak English as a second or third language. 

The use of the Figure with the screen shot of an excel spreadsheet has been very popular within our network of research collaborators. This image was chosen specifically to make the process as clear as possible, particularly for researchers who have a language barrier. 

Many of the figures have been redrawn and the abbreviations have been added to the images. 

Round 2
